# Believe What You See: Implicit Constraint Approach for Offline Multi-Agent Reinforcement Learning

**Yiqin Yang**[1][†], **Xiaoteng Ma**[1][†][‡] **Chenghao Li**[1], **Zewu Zheng**[1],
**Qiyuan Zhang**[2], **Gao Huang**[1], **Jun Yang**[1][‡], **Qianchuan Zhao**[1]
[1]Tsinghua University, [2]Harbin Institute of Technology
{yangyiqi19, ma-xt17, lich18}@mails.tsinghua.edu.cn, zzheng17@126.com,
zhangqiyuan19@hit.edu.cn, {gaohuang, yangjun603, zhaoqc}@tsinghua.edu.cn

## Abstract

Learning from datasets without interaction with environments (Offline Learning) is an essential step to apply Reinforcement Learning (RL) algorithms in real-world scenarios. However, compared with the *single-agent* counterpart, offline *multi-agent* RL introduces more agents with the larger state and action space, which is more challenging but attracts little attention. We demonstrate current offline RL algorithms are ineffective in multi-agent systems due to the accumulated extrapolation error. In this paper, we propose a novel offline RL algorithm, named *Implicit Constraint Q-learning* (ICQ), which effectively alleviates the extrapolation error by only trusting the state-action pairs given in the dataset for value estimation. Moreover, we extend ICQ to multi-agent tasks by decomposing the joint-policy under the implicit constraint. Experimental results demonstrate that the extrapolation error is successfully controlled within a reasonable range and insensitive to the number of agents. We further show that ICQ achieves the state-of-the-art performance in the challenging multi-agent offline tasks (StarCraft II). Our code is public online at https://github.com/YiqinYang/ICQ.

## 1 Introduction

Recently, reinforcement learning (RL), an active learning process, has achieved massive success in various domains ranging from strategy games [59] to recommendation systems [8]. However, applying RL to real-world scenarios poses practical challenges: interaction with the real world, such as autonomous driving, is usually expensive or risky. To solve these issues, offline RL is an excellent choice to deal with practical problems [3, 24, 35, 42, 15, 28, 4, 23, 54, 12], aiming at learning from a fixed dataset without interaction with environments.

The greatest obstacle of offline RL is the distribution shift issue [16], which leads to extrapolation error, a phenomenon in which unseen state-action pairs are erroneously estimated. Unlike the online setting, the inaccurate estimated values of unseen pairs cannot be corrected by interacting with the environment. Therefore, most off-policy RL algorithms fail in the offline tasks due to intractable overgeneralization. Modern offline methods (e.g., Batch-Constrained deep Q-learning (BCQ) [16]) aim to enforce the learned policy to be close to the behavior policy or suppress the $Q$-value directly. These methods have achieved massive success in challenging single-agent offline tasks like D4RL [14].

However, many decision processes in real-world scenarios belong to multi-agent systems, such as intelligent transportation systems [2], sensor networks [37], and power grids [7]. Compared with the single-agent counterpart, the multi-agent system has a much larger action space, which grows

---

[†]Equal Contribution.
[‡]Equal Corresponding.

35th Conference on Neural Information Processing Systems (NeurIPS 2021).

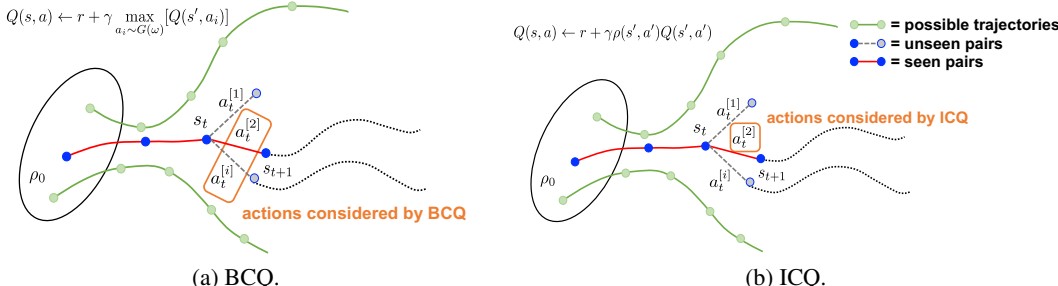

Figure 1: The comparison between ICQ and BCQ for the target $Q$-value estimation. The spots denote states, and the connections between spots indicate actions. The red solid-lines denote seen pairs, and the gray dotted-lines are unseen pairs. (a) BCQ estimates $Q$-value in a defined similar action set (orange) while unseen pairs still exist in the set with low probability. (b) ICQ only adopts seen pairs (orange) in the training set for $Q$-value estimation.

exponentially with the increasing of the agent number. When coming into the offline scenario, *the unseen state-action pairs will grow exponentially as the number of agents increases, accumulating the extrapolation error quickly*. The current offline algorithms are unsuccessful in multi-agent tasks even though they adopt the modern value-decomposition structure [26, 48, 25]. As shown in Figure 2, our results indicate that BCQ, a state-of-the-art offline algorithm, has divergent $Q$-estimates in a simple multi-agent MDP environment (e.g., BCQ (4 agents)). The extrapolation error for value estimation is accumulated quickly as the number of agents increases, significantly impairing the performance.

Based on these analyses, we propose the Implicit Constraint Q-learning (ICQ) algorithm, which effectively alleviates the extrapolation error as no unseen pairs are involved in estimating $Q$-value. Motivated by an implicit constraint optimization problem, ICQ adopts a SARSA-like approach [49] to evaluate $Q$-values and then converts the policy learning into a supervised regression problem. By decomposing the joint-policy under the implicit constraint, we extend ICQ to the multi-agent tasks successfully. To the best of our knowledge, our work is the first study analyzing and addressing the extrapolation error in multi-agent reinforcement learning.

We evaluate our algorithm on the challenging multi-agent offline tasks based on StarCraft II [40], where a large number of agents cooperatively complete a task. Experimental results show that ICQ can control the extrapolation error within a reasonable range under any number of agents and learn from complex multi-agent datasets. Further, we evaluate the single-agent version of ICQ in D4RL, a standard single-agent offline benchmark. The results demonstrate the generality of ICQ for a wide range of task scenarios, from single-agent to multi-agent, from discrete to continuous control.

## 2   Background

**Notation.** The fully cooperative multi-agent tasks are usually modeled as the Dec-POMDP [31] consisting of the tuple $G = \langle S, A, P, r, \Omega, O, n, \gamma \rangle$. Let $s \in S$ denote the true state of the environment. At each time step $t \in \mathbb{Z}^+$, each agent $i \in N \equiv \{1, \ldots, n\}$ chooses an action $a^i \in A$, forming a joint action $\boldsymbol{a} \in \mathbf{A} \equiv A^n$. Let $P(s' \mid s, \boldsymbol{a}) : S \times \mathbf{A} \times S \to [0, 1]$ denote the state transition function. All agents share the same reward function $r(s, \boldsymbol{a}) : S \times \mathbf{A} \to \mathbb{R}$.

We consider a partially observable scenario in which each agent draws individual observations $o^i \in \Omega$ according to the observation function $O(s, a) : S \times \mathbf{A} \to \Omega$. Each agent has an action-observation history $\tau^i \in \mathbf{T} \equiv (\Omega \times \mathbf{A})^t$, on which it conditions a stochastic policy $\pi^i(a^i \mid \tau^i)$ parameterized by $\theta_i : \mathbf{T} \times \mathbf{A} \to [0, 1]$. The joint action-value function is defined as $Q^{\boldsymbol{\pi}}(\boldsymbol{\tau}, \boldsymbol{a}) \triangleq \mathbb{E}_{s_{0:\infty}, \boldsymbol{a}_{0:\infty}} [\sum_{t=0}^{\infty} \gamma^t r_t \mid s_0 = s, \boldsymbol{a}_0 = \boldsymbol{a}, \boldsymbol{\pi}]$, where $\boldsymbol{\pi}$ is the joint-policy with parameters $\theta = \langle \theta_1, \ldots, \theta_n \rangle$. Let $\mathcal{B}$ denote the offline dataset, which contains trajectories of the behavior policy $\boldsymbol{\mu}$.

We adopt the *centralized training and decentralized execution* (CTDE) paradigm [43]. During training, the algorithm has access to the true state $s$ and every agent's action-observation history $\tau_i$,

as well as the freedom to share all information between agents. However, during execution, each agent has access only to its action-observation history.

**Batch-constrained deep Q-learning** (BCQ) is a state-of-the-art offline RL method, which aims to avoid selecting an unfamiliar action at the next state during a value update. Specifically, BCQ optimizes $\pi$ by introducing perturbation model $\xi(\tau, a, \Phi)$ and generative model $G(\tau; \varphi)$ as follows

$$\pi(\tau) = \underset{a^{[i]} + \xi(\tau, a^{[i]}, \Phi)}{\arg\max} \ Q^\pi(\tau, a^{[i]} + \xi(\tau, a^{[i]}, \Phi); \phi), \quad \text{s.t.} \quad \{a^{[i]} \sim G(\tau; \varphi)\}_{i=1}^m, \tag{1}$$

where $\pi$ selects the highest valued action from a collection of $m$ actions sampled from the generative model $G(\tau; \varphi)$, which aims to produce only previously seen actions. The perturbation model $\xi(\tau, a^{[i]}, \Phi)$ is adopted to adjust action $a^{[i]}$ in the range $[-\Phi, \Phi]$ to increase the diversity of actions.

# 3 Analysis of Accumulated Extrapolation Error in Multi-Agent RL

In this section, we theoretically analyze the extrapolation error propagation in offline RL, which lays the basis for Section 4. The extrapolation error mainly attributes the out-of-distribution (OOD) actions in the evaluation of $Q^\pi$ [16, 21]. To quantify the effect of OOD actions, we define the state-action pairs within the dataset as *seen* pairs. Otherwise, we name them as *unseen* pairs. We demonstrate that the extrapolation error propagation from the unseen pairs to the seen pairs is related to the size of the action space, which grows exponentially with the increasing number of agents. We further design a toy example to illustrate the inefficiency of current offline methods in multi-agent tasks.

## 3.1 Extrapolation Error Propagation in Offline RL

Following the analysis in BCQ [16], we define the tabular estimation error* as $\epsilon_{\text{MDP}}(\tau, a) \triangleq Q_M^\pi(\tau, a) - Q_B^\pi(\tau, a)$ (here we abuse $\tau$ to denote the state for analytical clarity), where the $M$ denotes the true MDP and $B$ denotes a new MDP computed from the batch by $P_B(\tau' \mid \tau, a) = \mathcal{N}(\tau, a, \tau') / \sum_{\tilde{\tau}} \mathcal{N}(\tau, a, \tilde{\tau})$. BCQ [16] has shown that $\epsilon_{\text{MDP}}(\tau, a)$ has a Bellman-like form with the extrapolation error $\epsilon_{\text{EXT}}(\tau, a)$ as the "reward function":

$$\epsilon_{\text{MDP}}(\tau, a) \triangleq \epsilon_{\text{EXT}}(\tau, a) + \sum_{\tau'} P_M(\tau' \mid \tau, a)\gamma \sum_{a'} \pi(a' \mid s')\epsilon_{\text{MDP}}(\tau', a'),$$

$$\epsilon_{\text{EXP}}(\tau, a) = \sum_{\tau'} \left(P_M(\tau' \mid \tau, a) - P_B(\tau' \mid \tau, a)\right)\left(r(\tau, a, \tau') + \gamma \sum_{a'} \pi(a' \mid \tau')Q_B^\pi(\tau', a')\right). \tag{2}$$

For the seen state-action pairs, $\epsilon_{\text{EXT}}(\tau, a) = 0$ since $P_M(\tau' \mid \tau, a) - P_B(\tau' \mid \tau, a) = 0$ in the deterministic environment. In contrast, the $\epsilon_{\text{EXT}}(\tau, a)$ of unseen pairs is uncontrollable and depends entirely on the initial values in tabular setting or the network generalization in DRL.

To further analyze how the extrapolation error in the unseen pairs impacts the estimation of actions in the dataset, we partition $\boldsymbol{\epsilon_{\text{MDP}}}$ and $\boldsymbol{\epsilon_{\text{EXT}}}$ as $\boldsymbol{\epsilon_{\text{MDP}}} = [\boldsymbol{\epsilon_s}, \boldsymbol{\epsilon_u}]^{\mathbf{T}}$ and $\boldsymbol{\epsilon_{\text{EXT}}} = [\mathbf{0}, \boldsymbol{\epsilon_b}]^{\text{T}}$ respectively according to seen and unseen state-action pairs. Let denote the transition matrix of the state-action pairs as $P_M^\pi(\tau', a' \mid \tau, a) = P_M(\tau' \mid \tau, a)\pi(a' \mid \tau')$. We decompose the transition matrix as $P_M^\pi = \left[P_{\text{s,s}}^\pi, P_{\text{s,u}}^\pi; P_{\text{u,s}}^\pi, P_{\text{u,u}}^\pi\right]$ according to state-action pairs' property (e.g., $P_{\text{s,u}}^\pi(\tau_{\text{u}}', a_{\text{u}}' \mid \tau_{\text{s}}, a_{\text{s}}) = P_M(\tau_{\text{u}}' \mid \tau_{\text{s}}, a_{\text{s}})\pi(a_{\text{u}}' \mid \tau_{\text{u}}')$ denotes the transition probability from seen to unseen pairs). Then the extrapolation error propagation can be described by the following linear system:

$$\begin{bmatrix} \boldsymbol{\epsilon_s} \\ \boldsymbol{\epsilon_u} \end{bmatrix} = \gamma \begin{bmatrix} P_{\text{s,s}}^\pi & P_{\text{s,u}}^\pi \\ P_{\text{u,s}}^\pi & P_{\text{u,u}}^\pi \end{bmatrix} \begin{bmatrix} \boldsymbol{\epsilon_s} \\ \boldsymbol{\epsilon_u} \end{bmatrix} + \begin{bmatrix} \mathbf{0} \\ \boldsymbol{\epsilon_b} \end{bmatrix}. \tag{3}$$

Based on the above definitions, we have the following conclusion.

**Theorem 1.** *Given a deterministic MDP, the propagation of $\boldsymbol{\epsilon_b}$ to $\boldsymbol{\epsilon_s}$ is proportional to $\|P_{\text{s,u}}^\pi\|_\infty$:*

$$\|\boldsymbol{\epsilon_s}\|_\infty \leq \frac{\gamma \left\|P_{\text{s,u}}^\pi\right\|_\infty}{(1 - \gamma)\left(1 - \gamma\left\|P_{\text{s,s}}^\pi\right\|_\infty\right)} \|\boldsymbol{\epsilon_b}\|_\infty. \tag{4}$$

---

*Note that we adopt a different definition of extrapolation error with BCQ. The $\epsilon_{\text{MDP}}(\tau, a)$ is regraded as the extrapolation error in BCQ, while the generalization error of unseen pairs $\epsilon_{\text{EXT}}(\tau, a)$ is considered in this work.

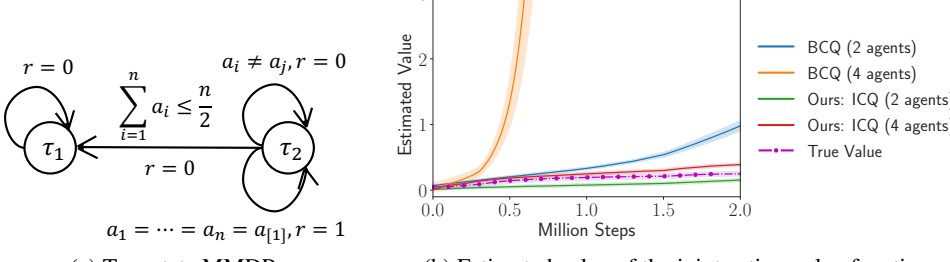

| (a) Two-state MMDP. | (b) Estimated value of the joint action-value function. |

Figure 2: (a) An MMDP where $Q$-estimates of BCQ will diverge as the number of agents increases. (b) The learning curve of the joint action-value function while running several agents in the given MMDP. The true values are similar in this task with different agent numbers, calculated by averaging the Monte-Carlo estimation under different agents. The $Q$-estimates of BCQ (4 agents) diverge while our algorithm (ICQ) has accurate $Q$-estimates. Please refer to Appendix C.2 for the complete results.

The above theorem indicates the effect of extrapolation error on seen state-action pairs is directly proportional to $\|P_{\mathrm{s,u}}^{\pi}\|_{\infty}$. In the practice, $\|P_{\mathrm{s,u}}^{\pi}\|_{\infty}$ is related to the size of action space and the dataset. If the action space is enormous, such as a multi-agent task with a number of agents, we need a larger amount of data to reduce $\|P_{\mathrm{s,u}}^{\pi}\|_{\infty}$. However, the dataset size in offline learning tasks is generally limited. Moreover, when using the networks to approximate the value function, $\epsilon_{\mathrm{b}}$ does not remain constant as $Q_{\mathcal{B}}(\tau_{\mathrm{u}}, a_{\mathrm{u}})$ could be arbitrary during training, making the $Q$-values extreme large even for the seen pairs. For these reasons, we have to enforce the $P_{\mathrm{s,u}}^{\pi} \to 0$ by avoiding using OOD actions. For example, BCQ utilizes an auxiliary generative model to constrain the target actions within a familiar action set (see Section 2 for a detailed description). However, the error propagation heavily depends on the accuracy of the generative model and is intolerable with the agent number increasing. We will demonstrate this effect in the following toy example.

### 3.2 Toy Example

We design a toy two states Multi-Agent Markov Decision Process (MMDP) to illustrate the accumulated extrapolation error in multi-agent tasks (see Figure 2a). All agents start at state $\tau_2$ and explore rewards for 100 environment steps by taking actions $a_{[1]} = 0$ or $a_{[2]} = 1$. The optimal policy is that all agents select $a_{[1]}$. The MMDP task has sparse rewards. The reward is 1 when following the optimal policy, otherwise, the reward is 0. The state $\tau_2$ will transfer to $\tau_1$ if the joint policy satisfies $\sum_{i=1}^{n} a_i \leq \frac{n}{2}$ at $\tau_2$, while the state $\tau_1$ will never return to $\tau_2$.

We run BCQ and our method ICQ on a limited dataset, which only contain 32 trajectories generated by QMIX. Obviously, the number of unseen state-action pairs exponentially grows as the number of agents increases. We control the amount of valuable trajectories ($r = 1$) in different datasets equal for fair comparisons. The multi-agent version of BCQ shares the same value-decomposition structure as ICQ (see Appendix D.2).

As shown in Figure 2b, the joint action-value function learned by BCQ gradually diverges as the number of agents increases while ICQ maintains a reasonable $Q$-value. The experimental result is consistent with Theorem 1, and we provide an additional analysis for the toy example in Appendix B.2. In summary, we show theoretically and empirically that the extrapolation error is accumulated quickly as the number of agents increases and makes the $Q$-estimates easier to diverge.

## 4 Implicit Constraint Approach for Offline Multi-Agent RL

In this section, we give an effective method to solve the accumulated extrapolation error in offline Multi-Agent RL based on the analysis of Section 3. From the implementation perspective, we find that a practical approach towards offline RL is to estimate target $Q$-value without sampled actions from the policy in training. We propose Implicit Constraint Q-learning (ICQ), which only trusts the seen state-action pairs in datasets for value estimation. Further, we extend ICQ to multi-agent tasks with a value decomposition framework and utilize a $\lambda$-return method to balance the variance and bias.

## 4.1 The Implicit Constraint Q-learning (ICQ) Approach

Based on the analysis of Section 3, we find that the extrapolation error can be effectively alleviated by enforcing the actions within the dataset when calculating the target values, which is the most significant difference between offline and off-policy RL. For a formal comparison of off-policy and offline algorithms, we first introduce the standard Bellman operator $\mathcal{T}^\pi$ as follows:

$$(\mathcal{T}^\pi Q)(\tau, a) \triangleq Q(\tau, a) + \mathbb{E}_{\tau'}[r + \gamma \mathbb{E}_{a' \sim \pi}[Q(\tau', a')] - Q(\tau, a)]. \tag{5}$$

Many off-policy evaluation methods, such as the Tree Backup [10] and Expected SARSA [41], are designed based on this operator. However, when coming into the offline setting, the standard Bellman operator suffers from the OOD issue as the actions sampled from current policy $\pi$ are adopted for target $Q$-value estimation. A natural way to avoid the OOD issue is adopting the importance sampling measure [30]:

$$(\mathcal{T}^\pi Q)(\tau, a) = Q(\tau, a) + \mathbb{E}_{\tau'}[r + \gamma \mathbb{E}_{a' \sim \mu}[\rho(\tau', a')Q(\tau', a')] - Q(\tau, a)], \tag{6}$$

where $\rho(\tau', a') \triangleq \frac{\pi(a'|\tau')}{\mu(a'|\tau')}$ denotes the importance sampling weight. If we can calculate $\rho(\tau', a')$ *with action $a'$ sampled from $\mu$ rather than $\pi$*, the unseen pairs will be avoided for target $Q$-value estimation. In this case, the extrapolation error is theoretically avoided since $P_{s,u}^\pi \to 0$. The estimated $Q$-value based on the above operation would be stable even in complex tasks with enormous action space. However, in most real-world scenarios, it is hard to obtain the exact behavior policy to calculate $\rho(\tau', a')$, e.g., using expert demonstrations. Fortunately, we find that the solution of following implicit constraint optimization problem is efficient to compute the desired importance sampling weight.

### 4.1.1 Implicit Constraint Q-learning

In offline tasks, the policies similar to the behavior policy are preferred while maximizing the accumulated reward $Q^\pi(\tau, a)$, i.e., $D_{\mathrm{KL}}(\pi \parallel \mu)[\tau] \leq \epsilon$. The policy optimization with the behavior regularized constraint can be described in the following problem:

$$\pi_{k+1} = \arg\max_\pi \mathbb{E}_{a \sim \pi(\cdot|\tau)}[Q^{\pi_k}(\tau, a)], \quad \text{s.t.} \quad D_{\mathrm{KL}}(\pi \parallel \mu)[\tau] \leq \epsilon. \tag{7}$$

This problem has well studied in many previous works [36, 1, 58]. Note that the objective is a linear function of the decision variables $\pi$ and all constraints are convex functions. Thus we can obtain the optimal policy $\pi^*$ related to $\mu$ through the KKT condition [9], for which the proof is in Appendix B.4:

$$\pi_{k+1}^*(a \mid \tau) = \frac{1}{Z(\tau)} \mu(a \mid \tau) \exp\left(\frac{Q^{\pi_k}(\tau, a)}{\alpha}\right), \tag{8}$$

where $\alpha > 0$ is the Lagrangian coefficient and $Z(\tau) = \sum_{\tilde{a}} \mu(\tilde{a} \mid \tau) \exp\left(\frac{1}{\alpha} Q^{\pi_k}(\tau, \tilde{a})\right)$ is the normalizing partition function. Next, we calculate the ratio between $\pi$ and $\mu$ by relocating $\mu$ to the left-hand side:

$$\rho(\tau, a) = \frac{\pi_{k+1}^*(a \mid \tau)}{\mu(a \mid \tau)} = \frac{1}{Z(\tau)} \exp\left(\frac{Q^{\pi_k}(\tau, a)}{\alpha}\right). \tag{9}$$

Motivated on Equation 9, we define the Implicit Constraint Q-learning operator as

$$\mathcal{T}_{\mathrm{ICQ}} Q(\tau, a) = r + \gamma \mathbb{E}_{a' \sim \mu} \left[\frac{1}{Z(\tau')} \exp\left(\frac{Q(\tau', a')}{\alpha}\right) Q(\tau', a')\right]. \tag{10}$$

Thus we obtain a SARAR-like algorithm which not uses any unseen pairs.

**Comparison with previous methods.** While BCQ learns an action generator to filter unseen pairs in $Q$-value estimation, it cannot work in enormous action space due to the error of the generator (see Figure 1). Instead, in the value update of ICQ, we do not use the sampled actions to compute the target values, thus we alleviate extrapolation error effectively. There are some previous works, such as AWAC [29] and AWR [35], addressing the offline problem with similar constrained problem in Equation 7. However, these methods only impose the constraint on the policy loss and adopt the standard Bellman operator to evaluate $Q$-function, which involves the unseen actions or converges to the value of behavior policy $\mu$. Differently, we re-weight the target $Q(\tau', a')$ with the importance sampling weight derived from the optimization problem, which makes the estimated value closer to the optimal value function.

### 4.1.2 Theoretical Analysis

The ICQ operator in Equation 10 results in a SARSA-like algorithm, which be re-written as:

$$\mathcal{T}_{\mathrm{ICQ}}Q(\tau,a) = r + \gamma \sum_{a'\in\mathcal{B}} \left[ \frac{1}{Z(\tau')}\mu(a'\mid\tau')\exp\left(\frac{1}{\alpha}Q(\tau',a')\right)Q(\tau',a')\right]. \tag{11}$$

This update rule can be viewed as a regularized softmax operator [46, 34] in the offline setting. When $\alpha \to \infty$, $\mathcal{T}_{\mathrm{ICQ}}$ approaches $\mathcal{T}^\mu$. When $\alpha \to 0$, $\mathcal{T}_{\mathrm{ICQ}}$ becomes the batch-constrained Bellman optimal operator $\mathcal{T}_{\mathrm{BCQ}}$ [16], which constrains the possible actions with respect to the batch:

$$\mathcal{T}_{\mathrm{BCQ}}Q(\tau,a) = r + \gamma \max_{a'\in\mathcal{B}} Q(\tau',a'). \tag{12}$$

$\mathcal{T}_{\mathrm{BCQ}}$ has been shown to converge to the optimal action-value function $Q^*$ of the batch, which means $\lim_{k\to\infty}\mathcal{T}_{\mathrm{BCQ}}^k Q_0 = Q^*$ for arbitrary $Q_0$. Based on this result, we show that iteratively applying $\mathcal{T}_{\mathrm{ICQ}}$ will result in a $Q$-function not far away from $Q^*$:

**Theorem 2.** *Let $\mathcal{T}_{\mathrm{ICQ}}^k Q_0$ denote that the operator $\mathcal{T}_{\mathrm{ICQ}}$ are iteratively applied over an initial state-action value function $Q_0$ for $k$ times. Then, we have $\forall(\tau,a)$, $\limsup_{k\to\infty}\mathcal{T}_{\mathrm{ICQ}}^k Q_0(\tau,a) \le Q^*(\tau,a)$,*

$$\liminf_{k\to\infty}\mathcal{T}_{\mathrm{ICQ}}^k Q_0(\tau,a) \ge Q^*(\tau,a) - \frac{\gamma(|A|-1)}{(1-\gamma)}\max\left\{\frac{1}{(\frac{1}{\alpha}+1)C+1}, \frac{2Q_{\max}}{1+C\exp(\frac{1}{\alpha})}\right\}, \tag{13}$$

*where $|A|$ is the action space, $|A_\tau|$ is the action space for state $\tau$, $C \triangleq \inf_{\tau\in S}\inf_{2\le i\le|A_\tau|}\frac{\mu(a_{[1]}|\tau)}{\mu(a_{[i]}|\tau)}$ and $\mu(a_{[1]}\mid\tau)$ denotes the probability of choosing the expert action according to behavioral policy $\mu$. Moreover, the upper bound of $\mathcal{T}_{\mathrm{BCQ}}^k Q_0 - \mathcal{T}_{\mathrm{ICQ}}^k Q_0$ decays exponentially fast in terms of $\alpha$.*

While $\mathcal{T}_{\mathrm{ICQ}}$ is not a contraction [5] (similar with the softmax operator), the $Q$-values are still within a reasonable range. Further, $\mathcal{T}_{\mathrm{ICQ}}$ converges to $\mathcal{T}_{\mathrm{BCQ}}$ with an exponential rate in terms of $\alpha$. Our result also quantifies the difficulty in offline RL problems. Based on the definition of $\mu(a_{[i]}|\tau)$, $C$ shows the proportion of the expert experience in the dataset. A larger $C$ corresponds to more expert experience, which induces a smaller distance between $\mathcal{T}_{\mathrm{ICQ}}^k Q_0(\tau,a)$ and $Q^*(\tau,a)$. In contrast, with a small $C$, the expert experience is few and the conservatism in learning is necessary.

### 4.1.3 Algorithm

Based on the derived operator $\mathcal{T}_{\mathrm{ICQ}}$ in Equation 9, we can learn $Q(\tau,a;\phi)$ by minimizing

$$\mathcal{J}_Q(\phi) = \mathbb{E}_{\tau,a,\tau',a'\sim\mathcal{B}}\left[r + \gamma\frac{1}{Z(\tau')}\exp\left(\frac{Q(\tau',a';\phi')}{\alpha}\right)Q(\tau',a';\phi') - Q(\tau,a;\phi)\right]^2, \tag{14}$$

where the $Q$-network and the target $Q$-network are parameterized by $\phi$ and $\phi'$ respectively.

As for the policy training, we project the non-parametric optimal policy $\pi_{k+1}^*$ in Equation 8 into the parameterized policy space $\theta$ by minimizing the following KL distance, which is implemented on the data distribution of the batch:

$$
\begin{aligned}
\mathcal{J}_\pi(\theta) &= \mathbb{E}_{\tau\sim\mathcal{B}}\left[D_{\mathrm{KL}}\left(\pi_{k+1}^*\|\pi_\theta\right)[\tau]\right] = \mathbb{E}_{\tau\sim\mathcal{B}}\left[-\sum_a \pi_{k+1}^*(a\mid\tau)\log\frac{\pi_\theta(a\mid\tau)}{\pi_{k+1}^*(a\mid\tau)}\right]\\
&\overset{(a)}{=} \mathbb{E}_{\tau\sim\mathcal{B}}\left[\sum_a \frac{\pi_{k+1}^*(a\mid\tau)}{\mu(a\mid\tau)}\mu(a\mid\tau)(-\log\pi_\theta(a\mid\tau))\right]\\
&\overset{(b)}{=} \mathbb{E}_{\tau,a\sim\mathcal{B}}\left[-\frac{1}{Z(\tau)}\log(\pi(a\mid\tau;\theta))\exp\left(\frac{Q(\tau,a)}{\alpha}\right)\right],
\end{aligned}
\tag{15}
$$

where $(a)$ ignores $\mathbb{E}_{\tau\sim\mathcal{B}}\left[\sum_a \pi_{k+1}^*(a\mid\tau)\log\pi_{k+1}^*(a\mid\tau)\right]$ that is not related to $\theta$, and $(b)$ applies the importance sampling weight derived in Equation 9 under forward KL constraint. Note that tuning the $\alpha$ parameter in Equation 15 between 0 and $\infty$ interpolates between $Q$-learning and behavioral cloning. See Appendix A for the complete workflow of the ICQ algorithm. We provide two implementation options to compute the normalizing partition function $Z(\tau)$, which is discussed in detail in Appendix D.1.

## 4.2 Extending ICQ to Multi-Agent Tasks

In the previous section, we propose an implicit constraint $Q$-learning framework by re-weighting target $Q$-value $Q(\tau', a')$ in the critic loss, which is efficient to alleviate the extrapolation error. We next extend ICQ to multi-agent tasks. For notational clarity, we name the **M**ulti- **A**gent version of ICQ as ICQ-MA.

### 4.2.1 Decomposed Multi-Agent Joint-Policy under Implicit Constraint

Under the CTDE framework, we have to train individual policies for decentralized execution. Besides, it is also challenging to compute $\mathbb{E}_{\boldsymbol{\mu}}[\rho(\boldsymbol{\tau}', \boldsymbol{a}')Q^{\boldsymbol{\pi}}(\boldsymbol{\tau}', \boldsymbol{a}')]$ in multi-agent policy evaluation as its computational complexity is $O(|A|^n)$. To address the above issues, we first define the joint-policy as $\boldsymbol{\pi}(\boldsymbol{a} \mid \boldsymbol{\tau}) \triangleq \Pi_{i \in N} \pi^i(a^i \mid \tau^i)$, and then introduce a mild value-decomposition assumption:

$$Q^{\boldsymbol{\pi}}(\boldsymbol{\tau}, \boldsymbol{a}) = \sum_i w^i(\boldsymbol{\tau})Q^i(\tau^i, a^i) + b(\boldsymbol{\tau}), \qquad (16)$$

where $w^i(\boldsymbol{\tau}) \geq 0$ and $b(\boldsymbol{\tau})$ are generated by the Mixer Network whose inputs are global observation-action history (see

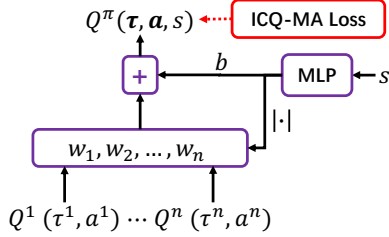

Figure 3: Mixer Network.

Figure 3). Based on the above assumptions, we propose the decomposed multi-agent joint-policy under implicit constraint in the following theorem:

**Theorem 3.** *Assuming the joint action-value function is linearly decomposed, we can decompose the multi-agent joint-policy under implicit constraint as follows*

$$\boldsymbol{\pi} = \arg\max_{\pi^1, ..., \pi^n} \sum_i \mathbb{E}_{\tau^i, a^i \sim \mathcal{B}} \left[ \frac{1}{Z^i(\tau^i)} \log(\pi^i(a^i \mid \tau^i)) \exp\left( \frac{w^i(\boldsymbol{\tau})Q^i(\tau^i, a^i)}{\alpha} \right) \right], \qquad (17)$$

*where $Z^i(\tau^i) = \sum_{\tilde{a}^i} \mu^i(\tilde{a}^i \mid \tau^i) \exp\left( \frac{1}{\alpha} w^i(\boldsymbol{\tau})Q^i(\tau^i, \tilde{a}^i) \right)$ is the normalizing partition function.*

The decomposed multi-agent joint-policy has a concise form. We can train individual policies $\pi^i$ by minimizing

$$\mathcal{J}_{\boldsymbol{\pi}}(\theta) = \sum_i \mathbb{E}_{\tau^i, a^i \sim \mathcal{B}} \left[ -\frac{1}{Z^i(\tau^i)} \log(\pi^i(a^i \mid \tau^i; \theta_i)) \exp\left( \frac{w^i(\boldsymbol{\tau})Q^i(\tau^i, a^i)}{\alpha} \right) \right]. \qquad (18)$$

Besides, $w^i(\boldsymbol{\tau})$ achieves the trade-off between the roles of agents. If some agents have important roles, the value of corresponding $w^i(\boldsymbol{\tau})$ is relatively large. Also, if $w^i(\boldsymbol{\tau}) \to 0$, $\pi^i$ is approximately considered as the behavior cloning policy. As for the policy evaluation, we train $Q(\boldsymbol{\tau}, \boldsymbol{a}; \phi, \psi)$ by minimizing

$$\mathcal{J}_Q(\phi, \psi) = \mathbb{E}_{\mathcal{B}} \left[ \sum_{t \geq 0} (\gamma\lambda)^t \left( r_t + \gamma \frac{1}{Z(\boldsymbol{\tau}_{t+1})} \exp\left( \frac{Q(\boldsymbol{\tau}_{t+1}, \boldsymbol{a}_{t+1})}{\alpha} \right) Q(\boldsymbol{\tau}_{t+1}, \boldsymbol{a}_{t+1}) - Q(\boldsymbol{\tau}_t, \boldsymbol{a}_t) \right) \right]^2, \qquad (19)$$

where $Q(\boldsymbol{\tau}_{t+1}, \boldsymbol{a}_{t+1}) = \sum_i w^i(\boldsymbol{\tau}_{t+1}; \psi')Q^i(\tau^i_{t+1}, a^i_{t+1}; \phi'_i) - b(\boldsymbol{\tau}_{t+1}; \psi')$.

### 4.2.2 Multi-Agent Value Estimation with $\lambda$-return

As the offline dataset contains complete behavior trajectories, it is natural to accelerate the convergence of ICQ with the $n$-step method. Here we adopt $Q(\lambda)$ [27] to improve the estimation of ICQ, which weights the future temporal difference signal with a decay sequence $\lambda^t$. Further, the constraint in Equation 7 implicitly meets the convergence condition of $Q(\lambda)$. Therefore, we extend the ICQ operator in Equation 10 to $n$-step estimation, which is similar to $Q(\lambda)$:

$$(\mathcal{T}^\lambda_{\text{ICQ}}Q)(\boldsymbol{\tau}, \boldsymbol{a}) \triangleq Q(\boldsymbol{\tau}, \boldsymbol{a}) + \mathbb{E}_{\boldsymbol{\mu}} \left[ \sum_{t \geq 0} (\gamma\lambda)^t \left( r_t + \gamma\rho(\boldsymbol{\tau}_{t+1}, \boldsymbol{a}_{t+1})Q(\boldsymbol{\tau}_{t+1}, \boldsymbol{a}_{t+1}) - Q(\boldsymbol{\tau}_t, \boldsymbol{a}_t) \right) \right], \qquad (20)$$

where $\rho(\boldsymbol{\tau}_t, \boldsymbol{a}_t) = \frac{1}{Z(\boldsymbol{\tau}_t)} \exp(\frac{1}{\alpha}Q(\boldsymbol{\tau}_t, \boldsymbol{a}_t))$ and hyper-parameter $0 \leq \lambda \leq 1$ provides the balance between bias and variance.

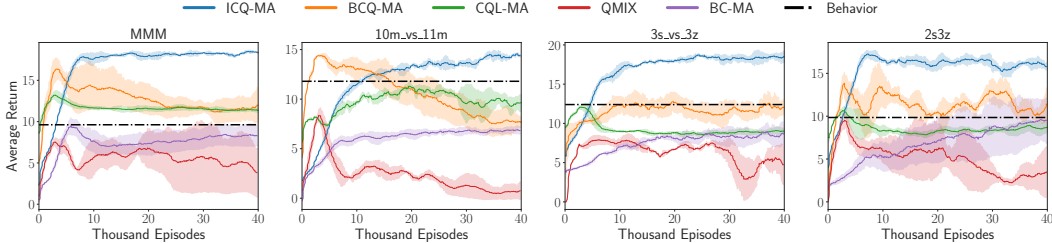

Figure 4: Performance comparison in offline StarCraft II tasks.

Table 1: Performance of ICQ with five offline RL baselines on the single-agent offline tasks with the normalized score metric proposed by D4RL benchmark [14], averaged over three random seeds with standard deviation. Scores roughly range from 0 to 100, where 0 corresponds to a random policy performance and 100 indicates an expert. The results for BC, BCQ, CQL, AWR and BRAC-p are taken from [14, 22].

| Dataset type | Environment | ICQ (ours) | BC | BCQ | CQL | AWR | BRAC-p |
|---|---|---|---|---|---|---|---|
| fixed | antmaze-umaze | **85.0 ± 2.7** | 65.0 | 78.9 | 74.0 | 56.0 | 50.0 |
| play | antmaze-medium | **80.0 ± 1.3** | 0.0 | 0.0 | 61.2 | 0.0 | 0.0 |
| play | antmaze-large | **51.0 ± 4.8** | 0.0 | 6.7 | 15.8 | 0.0 | 0.0 |
| diverse | antmaze-umaze | 65.0±3.3 | 55.0 | 55.0 | 84.0 | **70.3** | 40.0 |
| diverse | antmaze-medium | **65.0 ± 3.9** | 0.0 | 0.0 | 53.7 | 0.0 | 0.0 |
| diverse | antmaze-large | **44.0 ± 4.2** | 0.0 | 2.2 | 14.9 | 0.0 | 0.0 |
| expert | adroit-door | **103.9 ± 3.6** | 101.2 | 99.0 | - | 102.9 | -0.3 |
| expert | adroit-relocate | **109.5 ± 11.1** | 101.3 | 41.6 | - | 91.5 | -0.3 |
| expert | adroit-pen | **123.8 ± 22.1** | 85.1 | 114.9 | - | 111.0 | -3.5 |
| expert | adroit-hammer | **128.3 ± 2.5** | 125.6 | 107.2 | - | 39.0 | 0.3 |
| human | adroit-door | 6.4±2.4 | 0.5 | -0.0 | **9.1** | 0.4 | -0.3 |
| human | adroit-relocate | **1.5 ± 0.7** | -0.0 | -0.1 | 0.35 | -0.0 | -0.3 |
| human | adroit-pen | **91.3 ± 10.3** | 34.4 | 68.9 | 55.8 | 12.3 | 8.1 |
| human | adroit-hammer | 2.0±0.9 | 1.5 | 0.5 | **2.1** | 1.2 | 0.3 |
| medium | walker2d | 71.8±10.7 | 66.6 | 53.1 | **79.2** | 17.4 | 77.5 |
| medium | hopper | 55.6±5.7 | 49.0 | 54.5 | **58.0** | 35.9 | 32.7 |
| medium | halfcheetah | 42.5±1.3 | 36.1 | 40.7 | **44.4** | 37.4 | 43.8 |
| med-expert | walker2d | **98.9 ± 5.2** | 66.8 | 57.5 | 98.7 | 53.8 | 76.9 |
| med-expert | hopper | 109.0±13.6 | **111.9** | 110.9 | 111.0 | 27.1 | 1.9 |
| med-expert | halfcheetah | **110.3 ± 1.1** | 35.8 | 64.7 | 104.8 | 52.7 | 44.2 |

## 5 Related Work

As ICQ-MA seems to be the first work addressing the accumulated extrapolation error issue in offline MARL, we briefly review the prior single-agent offline RL works here, which can be divided into three categories: dynamic programming, model-based, and safe policy improvement methods.

**Dynamic Programming.** Policy constraint methods in dynamic programming [20, 3, 58, 51, 17] are most closely related to our work. They attempt to enforce $\pi$ to be close to $\mu$ under KL-divergence, Wasserstein distance [53], or MMD [47], and then only use actions sampled from $\pi$ in dynamic programming. For example, BCQ [16] constrains the mismatch between the state-action visitation of the policy and the state-action pairs contained in the batch by using a state-conditioned generative model to produce only previously seen actions. AWR [35] and ABM [42] attempt to estimate the value function of the behavior policy via Monte-Carlo or TD($\lambda$). Unlike these methods, our algorithm, ICQ, estimates the $Q$-function of the current policy using actions sampled from $\mu$, enabling much more efficient learning. Another series of methods [52, 32, 33] aim to estimate uncertainty to determine the trustworthiness of a $Q$-value prediction. However, the high-fidelity requirements for uncertainty estimates limit the performance of algorithms.

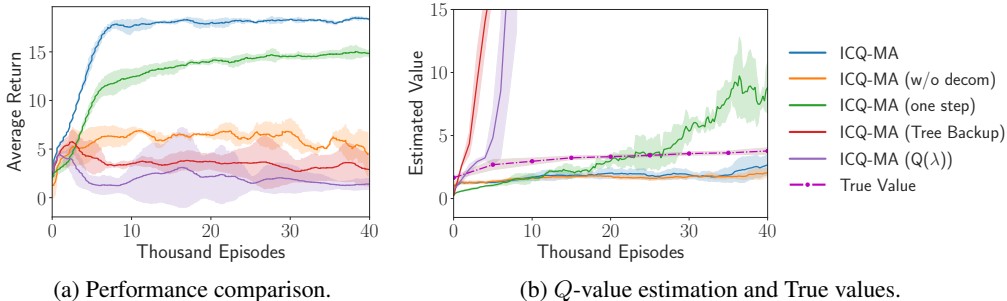

(a) Performance comparison.  (b) $Q$-value estimation and True values.

Figure 5: Module ablation study on MMM map.

**Model-based and Safe Policy Improvement.** Model-based methods [18, 50, 13, 56, 19] attempt to learn the model from offline data, with minimal modification to the algorithm. Nevertheless, modeling MDPs with very high-dimensional image observations and long horizons is a major open problem, which leads to limited algorithm performance [24]. Besides, safe policy improvement methods [23, 44, 6, 11] require a separately estimated model to $\mu$ to deal with unseen actions. However, accurately estimating $\mu$ is especially hard if the data come from multiple sources [29].

## 6 Experiments

In this section, we evaluate ICQ-MA and ICQ on multi-agent (StarCraft II) and single-agent (D4RL) offline benchmarks and compare them with state-of-the-art methods. Then, we conduct ablation studies on ICQ-MA. We aim to better understand each component's effect and further analyze the main driver for the performance improvement.

### 6.1 Multi-Agent Offline Tasks on StarCraft II

We first construct the multi-agent offline datasets based on ten maps in StarCraft II (see Table 2 in Appendix E). The datasets are made by collecting DOP [55] training data. All maps share the same reward function, and each map includes 3000 trajectories. We are interested in non-expert data or multi-source data. Therefore, we artificially divide behavior policies into three levels based on the average episode return (see Table 3 in Appendix E). Then, we evenly mix data of three levels.

We compare our method against QMIX [39], multi-agent version of BCQ (BCQ-MA), CQL (CQL-MA), and behavior cloning (BC-MA). To maintain consistency, BCQ-MA, CQL-MA, and BC-MA share the same linear value decomposition structure with ICQ-MA. Details for baseline implementations are in Appendix D.2. Each algorithm runs with five seeds, where the performance is evaluated ten times every 50 episodes. Details for hyper-parameters are in Appendix E.1.

We investigate ICQ-MA's performance compared to common baselines in different scenarios. Results in Figure 4 show that ICQ-MA significantly outperforms all baselines and achieves state-of-the-art performance in all maps. QMIX, BCQ-MA, and CQL-MA have poor performances due to the accumulated extrapolation error. Interestingly, since BC does not depend on the policy evaluation, it is not subject to extrapolation error. Thus BC-MA has a sound performance as StarCraft II is near deterministic. We implement BCQ and CQL according to their official code[*].

### 6.2 Single-Agent Offline Tasks on D4RL

To compare with current offline methods, we evaluate ICQ in the single offline tasks (e.g., D4RL), including gym domains, Adroit tasks [38] and AntMaze. Specifically, adroit tasks require controlling a 24-DoF robotic hand to imitate human behavior. AntMaze requires composing parts of sub-optimal trajectories to form more optimal policies for reaching goals on a MuJoco Ant robot. Experimental result in Table 1 shows that ICQ achieves the state-of-the-art performance in many tasks compared with the current offline methods.

---

[*]BCQ: https://github.com/sfujim/BCQ,   CQL: https://github.com/aviralkumar2907/CQL.

### 6.3 Ablation Study

We conduct ablation studies of ICQ-MA in the MMM map of StarCraft II to study the effect of different modules, value estimation, important hyper-parameters, and data quality.

**Module and Value Estimation Analysis.** From Figure 5, we find that if we adopt other $Q$-value estimation methods in implicit constraint policies (e.g., $Q(\lambda)$ [27] or Tree Backup), the corresponding algorithms (ICQ-MA ($Q(\lambda)$) or ICQ-MA (Tree Backup)) have poor performances and incorrect estimated values. Suppose we train ICQ-MA without decomposed implicit constraint module (e.g., ICQ-MA (w/o decom)). In that case, the algorithm's performance is poor, although the estimated value is smaller than the true value, confirming the necessity of decomposed policy. Besides, the performance of one-step estimation (ICQ-MA (one step)) indicates $n$-step estimation is not the critical factor for improving ICQ-MA, while one-step estimation will introduce more bias.

**The Parameter $\alpha$.** The Lagrangian coefficient $\alpha$ of implicit constraint operator directly affects the intensity of constraint, which is a critical parameter for the performance. A smaller $\alpha$ leads to a relaxing constraint and tends to maximize reward. If $\alpha \to 0$, ICQ-MA is simplified to $Q$-learning [57] while $\alpha \to \infty$ results in that ICQ-MA is equivalent to behavior cloning. Indeed, there is an intermediate value that performs best that can best provide the trade-off as in Appendix C.4.

**Data Quality.** It is also worth studying the performance of ICQ-MA and BC-MA with varying data quality. Specifically, we make the datasets from behavior policies of different levels (e.g., Good, Medium, and Poor). As shown in Figure 9 in Appendix C.4, ICQ-MA is not sensitive to the data quality, while the performance of BC-MA drops drastically with the data quality deteriorates. Results confirm that ICQ-MA is robust to the data quality while BC-MA strongly relies on the data quality.

**Computational Complexity.** With the same training steps in SMAC, BCQ-MA consumes 70% time of ICQ-MA. Although ICQ-MA takes a little long time compared with BCQ-MA, it achieves excellent performance in benchmarks. The computing infrastructure for running experiments is a server with an AMD EPYC 7702 64-Core Processor CPU.

## 7  Conclusion

In this work, we demonstrate a critical problem in multi-agent off-policy reinforcement learning with finite data, where it introduces accumulated extrapolation error in the number of agents. We empirically show the current offline algorithms are ineffective in the multi-agent offline setting. Therefore, we propose the Implicit Constraint Q-learning (ICQ) method, which effectively alleviates extrapolation error by only trusting the state-action pairs in datasets. To the best of our knowledge, the multi-agent version of ICQ is the first multi-agent offline algorithm capable of learning from complex multi-agent datasets. Due to the importance of offline tasks and multi-agent systems, we sincerely hope our algorithms can be a solid foothold for applying RL to practical applications.

## Acknowledgments and Disclosure of Funding

This work was funded by the National Natural Science Foundation of China (ID:U1813216), National Key Research and Development Project of China under Grant 2017YFC0704100 and Grant 2016YFB0901900, in part by the National Natural Science Foundation of China under Grant 61425027, the 111 International Collaboration Program of China under Grant BP2018006, and BNRist Program (BNR2019TD01009) and the National Innovation Center of High Speed Train R&D project (CX/KJ-2020-0006).

We sincerely appreciate reviewers, whose valuable comments have benefited our paper significantly!

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
