# OpenReview forum: "Believe What You See: Implicit Constraint Approach for Offline Multi-Agent Reinforcement Learning"
_NeurIPS.cc/2021/Conference — NeurIPS 2021 Spotlight_

### Official Review · Reviewer_xr9t · 2021-06-28

**Rating:** 6
**Confidence:** 5

**Summary:**

This paper studies offline RL in multi-agent setting. The analysis shows that when the number of agents increases, the extrapolation error would increase. To alleviate this issue, the authors propose a method called Implicit Contraint Q learning (ICQ) that learns Q value while contraining the policy to be close to what have been seen in the dataset. Experimental results show that in MARL setting, the proposed method outperforms baselines such as MA-CQL and MA-BCQ. In the single-agent setting, the results seem comparable to the previous method.

**Limitations And Societal Impact:**

Experimental results seem a bit unconvincing for CQL is significantly lower than BCQ in D4RL envs. This is not consistent with the reported results in the original paper. If baselines are not well implemented, it weakens the quality of the paper directly. Can you compare your results with the original paper and show they are comparable (no need to be the same)?

**Main Review:**

The paper follows a popular intuition "conservatism" in this area. While this general idea has been explored many times, they are the first to work in multi-agent setting.

The paper is not well-written.  I believe they did not differentiate or clarify their key difference from past work such as CQL and MOPO [2]. Both of these works are doing "believe what you have seen" to some extend. Why do they fail in MARL setting but ICQ succeeds?
The toy example is great for understanding the error. But a comparison between ICQ and CQL would be great on that.

The lack of improvements in the D4RL setting is surprising. A more detailed analysis should be made and some qualitative evidence can be useful for understanding the difference between D4RL and your MA environments.

Can you specify the reason for the MA environments? There are multiple options from starcraft minigames.

I think I would consider changing the score if you could run baselines properly and show that they match the performance in the original paper.

Many typos such as "find-tuned" need to be adjusted.


[1] Kumar, Aviral, et al. "Conservative q-learning for offline reinforcement learning." arXiv preprint arXiv:2006.04779 (2020).
[2] Yu, Tianhe, et al. "Mopo: Model-based offline policy optimization." arXiv preprint arXiv:2005.13239 (2020).


**Time Spent Reviewing:**

2

---

> ### Author Response · Authors · 2021-08-10
> **Response to Reviewer xr9t**
>
> Dear Reviewer,
>
> Thanks for your valuable comments.
> We hope the following statement can address your concern. We add the supplementary experimental results on single-agent offline tasks as part of our response.
>
> **Q1: The lack of clarity of the difference between ICQ and past work such as CQL and MOPO.**
>
> A1: Although CQL and MOPO are doing "believe what you have seen" to some extent, there are still significant differences between ICQ, CQL, and MOPO. The significant difference is that ICQ aims to avoid unseen pairs in policy evaluation by re-weighting the target Q using an importance sampling ratio. The importance sampling ratio is derived by maximizing Q-values, subject to a KL-constraint between current policy $\pi$ and the behavior policy $\mu$. Therefore, ICQ avoids an additional regularization term as CQL does or even training a separate network as BCQ does, which contributes ICQ to be a simple yet efficient method. Also, based on the analysis of Theorem 1 and Theorem 2, ICQ can eliminate the extrapolation error to zero.
>
> Although CQL gives a theoretical guarantee for the Q-estimation, CQL has to balance the regularization term with other optimization goals, which makes the performance of CQL unstable.
>
> As pointed in the original paper of MOPO, the practical uncertainty-penalized reward of MOPO is computed based on the maximum standard deviation of the learned models in the ensemble, which lacks theoretical guarantees and may be unsafe due to the model error.
>
> We admire the problem formulation, insights in CQL and MOPO very much. We do think their exploration of offline RL is groundbreaking. We have put the comparison and discussion with CQL and MOPO in the critical place of the paper.
>
>
> **Q2: Why CQL seems to be not consistent with the reported results in the original paper?**
>
> A2:  We used the code provided by the author in Github without any modification in D4RL tasks.
> However, due to sensitivity to hyperparameters, CQL has unstable performances.
>
> To clear your confusion, we re-compare ICQ with CQL in D4RL tasks, where the results for CQL are taken from the original paper. Please refer to the supplementary experimental results. ICQ still works better in complex single-agent offline tasks than CQL, such as antmaze-medium/large, door/hammer-human. We appreciate that you pointed out the confusion for experiments on D4RL, and we have replaced the results of CQL on D4RL with reported results in the CQL paper.
>
> **Q3: The insufficient experiments on D4RL tasks.**
>
> A3:
> We only test ICQ on simple D4RL tasks in the paper. Therefore, it is not surprising that ICQ has a slight improvement compared with BCQ.
>
> In additional experimental results, we test ICQ with six offline SOTA methods on complex D4RL tasks. ICQ achieves state-of-the-art performance in complex offline RL tasks, such as antmaze-medium/large. We will put the additional experimental results on the updated paper to make the experiments more convincing.
>
> **Q4: The lack of detailed analysis and  qualitative evidence between D4RL and MA environments.**
>
> A4: D4RL are built on the single-agent simulator such as Mujoco and Adroit. Compared with single-agent tasks on D4RL, the simulated StarCraft II environment and carefully designed scenarios require learning-rich cooperative behaviors under partial observability, which is challenging. Therefore, the MA environments require more complex controlling on the independent agents.
> Specifically, in the MMM map, we need to control ten agents. Each agent has 160-dimensional observations and 16-dimensional actions. More complex maps have more agents to control. However, D4RL tasks only contain one agent. Moreover, the adroit-pen task has 45-dimensional observations and 24-dimensional actions. The mujoco-halfcheetah task has 17-dimensional observations and 6-dimensional actions.
> Therefore, the unseen pairs in StarCraft II environment are much more than D4RL tasks.
>
> **Q5: The lack of explanation for choosing starcraft minigames.**
>
> A5: We used ten maps to cover most of the battle scenes as much as possible. For example, we adopt six easy maps, two hard maps, and two super-hard maps, MMM2 and 27m-vs-30m, including isomorphic agents, heterogeneous agents, and various types of combat missions. The number of agents varies in these maps. We appreciate that you pointed out the lack of explanation for the experiment, and we have updated our draft accordingly.
>
> We sincerely thank the reviewer again for the timely and valuable comments.  We hope that our response and additional experimental results have cleared most of your concerns. We are also always open to any further suggestions that may help improve our paper in the final version.

---

> > ### Comment · Reviewer_xr9t · 2021-08-19
> > **Rebuttal**
> >
> > The authors addressed my concern. I will raise my score to 6.

---

> > > ### Author Response · Authors · 2021-08-20
> > > **Thanks for raising the score!**
> > >
> > > We really appreciate the feedback from the reviewer and thanks for raising the score!

---

> ### Author Response · Authors · 2021-08-10
> **Looking forward to further discussions!**
>
> Dear reviewer,
>
> We were wondering if our response and revision have cleared all your concerns. In the previous responses, we have tried to address the points you have raised. In the remaining 1 day of the rebuttal period, we would appreciate it if you could re-evaluate our submission, or kindly let us know whether you have any other questions, so that we can still have time to respond and address them. We are looking forward to discussions that can further improve our current manuscript. Thanks!
>
> Best regards,
>
> The Authors

---

### Official Review · Reviewer_Rr7B · 2021-06-30

**Rating:** 5
**Confidence:** 3

**Summary:**

This work addresses the problem of reducing the error incurred by unseen state-action pairs in offline reinforcement learning. They additionally show that in multiagent reinforcement learning settings this error grows exponentially in the number of agents in the system. To remedy this problem, they introduce a new algorithm Implicit Constraint Q-Learning (ICQ) that modifies the expected Bellman update with importance sampling to remove the dependence on the current policy and behaviour policy. The update now need not consider actions not existing in the dataset that were previously considered when calculating the importance sampling ratio. The authors demonstrate this method in both the single- and multi-agent scenario, and include a brief ablation study.

**Ethical Concerns:**

No.

**Limitations And Societal Impact:**

I suggest the authors include a discussion about the impact of the large behavioural bias that is introduced as a result of trajectories mimicked in the offline dataset. Additional discussion as to how ICQ might influence this problem would be helpful.

**Main Review:**

This work tackles an important problem in reinforcement learning: how to learn in the offline reinforcement learning setting. The key insight in this work is to separate the controllable error, resulting from the seen state-action pairs existing in the dataset, from the uncontrollable error, resulting from unseen state-action pairs. They provide an interesting theoretical analysis of this insight and its implications in Q-learning (Thm 1).

However, past this point the manuscript does not seem to have a straightforward story to follow. I believe this is due to the fact that the paper motivates itself with an issue particular to multiagent RL (combinatorial explosion of unseen state-action pairs of data growing in the number of agents) but does not directly address this issue. Instead, focuses on the related single-agent RL problem of a single agent’s unseen state-action pairs. This subtle distinction in problem statements forgoes consideration of the increasing number of other-agents, nor considers methods that can leverage known components of the game to reduce said complexity (e.g., consider roles for agents, we can now reuse unseen state-action pairs within agents of the same role). As a result of trying to build a narrative that misses this point, the authors need to introduce ideas in a counter-intuitive ordering that is difficult to follow.

In regards to their previous ICML reviews:
The inclusion of the code in the submission improves the ability to reproduce their results. The paper itself is still not detailed enough to reimplement their performance. I would encourage the authors also to include their offline datasets and saved final policies in the code release.
The inclusion of the ablation study in Sec 6.3 greatly enhances the understanding of the contributions of each modification to the downstream performance.
Single-agent tasks results are provided but are effectively a footnote and not discussed.

In summary, I think there are several interesting ideas within this paper. Despite this, I think the primary focus of this work should be on the single-agent setting so that the main contribution of this paper, ICQ, can be studied in more depth. The extension of ICQ into multiagent settings should be left as a follow-up work, and focus study on modifications that incorporate the multiagent aspect of the problem.


Major Comments (affecting review score):

 - The organization of the paper is difficult to follow and should be revised. Examples: (a) Sec 3.2 introduces a toy example studying the differences between BCQ and ICQ however ICQ is not introduced at all at this point. (b) Sec 4.2, the need for the two new proposed extensions is not clearly motivated; moreover, Sec 4.2.1 appears again to be a single-agent algorithm instead of a multiagent algorithm. (c) the experimental results do not follow the logical order of complexity that was introduced in the methodology section.

 - Single-agent claimed results do not follow from the provided results and are not discussed. Sec 6.2 claims a comparison that across a large variety of offline tasks (L269-270) in the single-agent setting ICQ significantly outperforms baselines. Despite this, only four environments results are shown in Figure 5, and the claimed result difference is not demonstrated as statistically significant nor does it appear to be a large (or small in some cases) improvement upon inspection of the graphs. The authors should either remove the claim of significance or provide the corresponding statistical tests. Moreover, more analysis in the single-agent setting would be beneficial. Perhaps providing breakdowns across seen and unseen transitions?

 - Unsupported claims and lacking baselines in multi-agent tasks. The dataset that the offline agents are trained with is generated from DOP, this algorithm’s performance should be included in the results. This provides a baseline for the performance of independent RL and contextualizes the quality of the behaviour cloning algorithms. It is surprising to see that a suite of different baselines are all underperforming compared to behaviour cloning. This raises suspicion of the quality of the reproduction of these algorithms. Do the authors reuse the same codebase as the source algorithms and match the performance they report on their experiments? Currently the authors claim this is a result of the extrapolation error; however, the reward curves provided offer no indication as to the source of the error. Additional analysis is necessary to support this claim.

Minor Comments (not affecting review score):

- It would be advantageous if the authors could formally relate their ICQ algorithm’s error (Sec 4.1) with the error components introduced in Sec 3.

- Sec 2, L73: it would be helpful to explain the perturbation model and its connection to ICQ.

- Sec 3, L94-96: it’s not clear how e_u and e_b are able to be disentangled. Additional discussion would be helpful.

- Sec 3.2, L122: “The optimal policy is all agents…”, this is not the optimal policy, but a particular equilibrium. Each agent’s optimal policy is to select A(1).

- Sec 4.1, L145: Have you considered using a greedy Bellman update as opposed to the expected Bellman update?


**Time Spent Reviewing:**

8

---

> ### Author Response · Authors · 2021-08-10
> **Response to Reviewer Rr7B**
>
> Dear Reviewer,
>
> Thanks for finding our work important and theoretical results interesting. The main issues regarding clarity and experimental results are addressed as follows. We add the supplementary experimental results on single-agent offline tasks as part of our response.
>
> **Q1:(Major Comments) The organization of the paper is difficult to follow.**
>
> A1: We apologize for your hard reading, while we respectfully disagree that the example supplied of you for two reasons:
>
> 1) We apply the toy example in Section 3.2 to demonstrate empirically the Theorem 1 and Theorem 2.
> Specifically, Theorem 1 demonstrates the extrapolation error propagation in single-agent offline tasks. Theorem 2 demonstrates the extrapolation error is accumulated with the growth of numbers of agents. Then, based on the analysis of Section 3 we propose ICQ and ICQ-MA in Section 4.
>
> 2) In Sec 4.2, we propose adopting the n-step estimation to compromise between bias and variance based on Theorem 2. Specifically, although ICQ-MA has eliminated the extrapolation error, the bias caused by the neural network can also affect the performance in multi-agent offline tasks. That's why we use n-stem estimation, which has been demonstrated empirically in Figure 6 in the paper. Moreover, the title of Section 4.2 is **Extending ICQ to Multi-Agent Tasks** and the title of Section 4.2.1 is **Multi-Agent Policy Evaluation with $\lambda$-return**. There may be some misunderstandings here.
>
> **Q2:(Major Comments) Single-agent claimed results do not follow from the provided results and are not discussed.**
>
> A2: We used the code provided by the author in Github without any modification in D4RL tasks.
> However, due to sensitivity to hyperparameters, CQL has very unstable performances.
>
> To clear your confusion, we re-compare ICQ with CQL in D4RL tasks, where the results for CQL are taken from the original paper. Please refer to the supplementary experimental results. ICQ still works better in complex single-agent offline tasks than CQL, such as antmaze-medium/large, door/hammer-human.
>
> **Q3:(Major Comments)Unsupported claims and lacking baselines in multi-agent tasks.**
>
> A3:
> As DOP does not have any techniques to deal with the extrapolation error, we did not add DOP to the baselines.
>
> It is not surprising that behavioral cloning is a reasonable choice with good enough offline data, i.e., collected by human experts. As reported in D4RL, BC beats many offline baselines in some complex single-agent offline tasks. For instance, BC achieves 101.3 in relocate-expert, while BCQ only has 41.6. Apart from the factor of data quality, one reason making BC a strong baseline is that **it never suffers from extrapolation error since it does not evaluate the value!** Instead, in spite of sound performances in some tasks, the current offline algorithms are still not good enough in handling the extrapolation error well. In our experiment, Q estimates of BCQ and BEAR explode in complex single-agent offline tasks, such as relocate/door-human. Based on our analysis in Section 3, multi-agent offline tasks are less tolerant of extrapolation errors than single-agent offline tasks.
> Therefore, most single-agent offline RL methods cannot be extended in multi-agent offline tasks.
>
> Thanks again for the valuable comments.
> We believe there is some misunderstanding here, and we sincerely hope our additional experimental results and response have cleared the concern so that the reviewer can re-evaluate our paper.  More comments on further improving the presentation are also very much welcomed.

---

> > ### Comment · Reviewer_Rr7B · 2021-08-10
> > **Response**
> >
> > Thank you for taking the time to address some of my concerns.
> >
> > Q1: This does help clarify the choice of organization for this aspect of the work. However, I used it only as an example, and to me the organization increases the difficulty of understanding (at least for me). This may be a stylistic component as not all reviewers made similar comments.
> >
> > Q2: I find baseline metrics that are copied from the original works to be highly suspicious. As we know, RL is a very sensitive and finicky problem domain, so any minor changes in the environment or experimental design between the algorithms can dramatically influence performance. It would be advantageous for the authors to reproduce the baselines in the same experimental condition, appropriately tune each algorithm for the differing condition, and provide error bounds for them.
> >
> > Q3: That is a good insight regarding BC. If that is not already in the paper I would suggest including this in your analysis to strengthen your argument. It is indeed intuitive that error compounds more dramatically when transition from single- to multi-agent; however, I disagree that these methods cannot be extended to multi-agent tasks. The task of generating a best-response policy on a fixed opponent policy, in the case of offline marl, is in fact a single-agent task (where the "environment" is the union of the environment and opponent strategy). The difference is that there is a known structure to the environment and methods are seeking to bake-in priors to exploit such structure (this work).
> >
> >  I further concur with Reviewer 5dG2's observation regarding dataset quality.

---

> > > ### Author Response · Authors · 2021-08-11
> > > **Response to your further discussion.**
> > >
> > > Dear Reviewer,
> > >
> > > Thanks for your detailed and valuable comments.
> > > We hope the following response can address your remaining concerns.
> > > We add supplementary experimental results on multi-agent offline tasks as part of our response.
> > >
> > > **Q1:(Major Comments) The organization of the paper is difficult to follow.**
> > >
> > > We apologize for the vagueness of the organization.
> > > We agree with your suggestion that we should focus on the single-agent setting as our main contribution is ICQ.
> > > Therefore, we re-organize our paper as follows:
> > >
> > > - **Title**: As the essential contribution of ICQ is its simple yet efficient extrapolation error control method, we respectfully ask your opinion whether our Title can be changed to:
> > >
> > > > A Simple Yet Efficient Extrapolation Error Control Method and its Application into Multi-Agent Systems.
> > >
> > > - **Organization**:
> > > As suggested, the new version of the paper focuses on the single-agent offline algorithm.
> > > We move the multi-agent version of ICQ into the appendix as the application in specific scenarios.
> > > In the experimental section, we add the additional experiments and extrapolation error control analysis on D4RL tasks.
> > > (Please refer to the updated supplementary experimental results in General remark.)
> > > ICQ has low estimation error even in complex tasks (e.g., random and human).
> > >
> > > **Q2:(Major Comments) Reproduce the baselines in the same experimental condition.**
> > >
> > > We totally agree that reproducing the baselines in the same experimental condition.
> > > We have tried to do it.
> > > However, some hyper-parameters are missing in the original papers, which induces difficulties in fully reproducing these baselines.
> > > As most offline RL algorithms are tested in the same tasks on D4RL, we apply the general comparison method in current offline RL papers [1-3] to ensure fairness.
> > > Therefore, we copy the baseline metrics from the original works, which overcomes the limitations of reproducing algorithms.
> > >
> > > **Q3:(Major Comments) Whether current offline RL methods can be extended to multi-agents tasks?**
> > >
> > > Thanks for finding our insights regarding BC is good, and we have added these analyzes in the paper.
> > > We agree that offline marl is, in fact, a single-agent task as we could consider the multi-agent tasks to be a re-description of single-agent tasks at the system level.
> > > However, in engineering, the complexity of multi-agents is much higher than that of single-agents.
> > > Specifically, in the MMM map in the StarCraft II environment, we need to control ten agents.
> > > Each agent has 16-dimensional actions.
> > > In contrast, D4RL tasks only contain one agent, which has a smaller action space.
> > > Therefore, as an application scenario, multi-agent tasks have specific engineering research value.
> > > We will explain the significant difference between offline multi-agent and single-agent RL as follows, while we will move these analyzes to the appendix as the paper will focus on the single-agent setting.
> > >
> > > -------
> > >
> > > It is necessary to point out that the size of action space is important in offline problems because the extrapolation error mainly comes from the target value calculating, which is $\max_b Q(s^\prime, b)$ in standard Bellman operator (see Figure 2 for a visual interpretation).
> > > This is exactly the biggest difference between the multi-agent and the single-agent problems: *the joint action space will grow exponentially as the number of agents increases in multi-agent systems*.
> > > These two evidences, extrapolation error getting larger with large action space and the action space getting larger with agent number increasing, together explain why ICQ is suitable for the multi-agent problem.
> > >
> > > To better demonstrate the above analysis, we report the performances with different parameters of BCQ-MA on multi-agent offline tasks in the following table.
> > > $\zeta$ in BCQ-MA denotes the conservative coefficient.
> > > Note that the performance of ICQ-MA is 18.85, and the averaged estimation value on the full trajectory is 3.17.
> > > The averaged estimation error is -0.11, which shows that the estimation value is close to the true value.
> > >
> > > Table 1. Performance of BCQ-MA on the multi-agent offline task (MMM map). Performance scores range from 0 to 20.
> > > Averaged value refers to the average estimated Q-values on the full trajectory.
> > > Averaged estimation error refers to the averaged value minus the average returns.
> > >
> > > | BCQ-MA | $\zeta$=0.1 | $\zeta$=0.3 | $\zeta$=0.5 | $\zeta$=0.7 | $\zeta$=0.9 |
> > > | --- | :---: | :---: | :---: | :---: | :---: |
> > > | Performance  | 9.126 | 10.173 | 10.091 | 9.623 | 9.14  |
> > > | Averaged Value | 261.1  | 102.8  | 92.19  | 91.19  | 38.51 |
> > > | Averaged Estimation Error | 259.83 | 101.53 | 90.92 | 89.92 | 37.24 |
> > >
> > > Based on our experiments, BCQ-MA becomes more conservative with $\zeta$ being larger, while the overall Q values are overestimated (even with $\zeta=0.9$).
> > > Therefore, significant overestimation induces poor performance in multi-agent offline tasks.
> > >
> > > **Q4:(Minor Comments) The relationship between the ICQ algorithm's error and the error components introduced in Section 3.**
> > >
> > > In section 3, we analyze the extrapolation error propagation in offline tasks.
> > > We propose ICQ in section 4.1 to avoid using unseen pairs to eliminate the extrapolation error based on these analyses.
> > > The remaining error of ICQ is bias error caused by the neural network.
> > >
> > > **Q5:(Minor Comments) The relationship between the perturbation model and ICQ.**
> > >
> > > Both BCQ and ICQ aim at avoiding the extrapolation error. The tradeoff is that BCQ needs to tolerate the additional error causing by the perturbation model, while ICQ adopts the implicit constraint.
> > > BCQ may be the first choice when we have an accurate behavior policy model or tabular case.
> > > However, ICQ has a greater tolerance for different tasks, from single-agent to multi-agent tasks, from discrete to continuous spaces.
> > >
> > > **Q6:(Minor Comments) How $e\_u$ and $e\_b$ are able to be disentangled?**
> > >
> > > The extrapolation error $e\_u$ mainly comes from neural network generalization, since it may be impossible to learn an action-value function with actions not contained in the dataset.
> > > We assume the unseen pairs have uncontrollable estimation errors caused by neural network generalization.
> > > Thus, we use $e\_b$ to simulate the function approximation error in a tabular setting. The assumption is reasonable as the overestimated unseen pairs are easy to exploit, especially with the maximization operation.
> > > We will add a detailed explanation of $e\_u$ and $e\_b$ in section 3.1 to make the paper clearer.
> > >
> > > **Q7:(Minor Comments) Unsuitable statement.**
> > >
> > > Thank you for pointing out the unsuitable presentation and providing detailed instructions. We have improved the current version as per your suggestions.
> > >
> > > **Q8:(Minor Comments) The discussion about the greedy Bellman update.**
> > >
> > > We ever considered the greedy Bellman update in l145, which means we can learn the optimal value function $Q^* $ by applying some techniques to deal with the extrapolation error.
> > > The agent makes decisions based on the learned $Q^* $.
> > > This is a good insight, and we want to explore it further as our future work.
> > >
> > > **Q8:(Minor Comments) Release code and offline datasets.**:
> > >
> > > All the source code including the baselines and offline datasets will be published in Github for reproducibility.
> > >
> > > Thanks again for your detailed comments.
> > > We sincerely hope our response has cleared your concerns, and we are looking forward to more discussions.
> > >
> > > [1] Kostrikov, Ilya, et al. "Offline reinforcement learning with fisher divergence critic regularization." International Conference on Machine Learning. PMLR, 2021.
> > >
> > > [2] Dadashi, Robert, et al. "Offline Reinforcement Learning with Pseudometric Learning." arXiv preprint arXiv:2103.01948 (2021).
> > >
> > > [3] Wu, Yue, et al. "Uncertainty Weighted Actor-Critic for Offline Reinforcement Learning." arXiv preprint arXiv:2105.08140 (2021).

---

> ### Author Response · Authors · 2021-08-20
> **Looking forward to further comments!**
>
> Dear reviewer,
>
> We have updated our supplementary experimental results in single-agent offline tasks by adding the estimation error of ICQ. We also updated our response based on the additional experimental results in multi-agent offline tasks. We are wondering if our response and revision have cleared your concerns.
> We would appreciate it if you could kindly let us know whether you have any other questions. We are looking forward to comments that can further improve our current manuscript. Thanks!
>
> Best regards,
>
> The Authors

---

> > ### Comment · Reviewer_Rr7B · 2021-08-20
> > **Response**
> >
> > Thank you for replying to all of my major and minor comments. I've increased my score in response to the increase in the quality of the submission.

---

> > > ### Author Response · Authors · 2021-08-20
> > > **Thanks for raising the score!**
> > >
> > > We appreciate the reviewer for raising the score! Thanks for the valuable comments and suggestions!

---

### Official Review · Reviewer_5dG2 · 2021-07-13

**Rating:** 5
**Confidence:** 4

**Summary:**

The paper proposes ICQ, a batch RL method which is a modification of BCQ with an update rule that uses only the state-action pairs present in the dataset to fight overestimation error. It can be seen as performing constrained optimization to keep the learned policy close to the behavior policy in KL space.

The authors use their method in the multi-agent context, and show that as the number of agents grows, the extrapolation error grows exponentially if one doesn't control transition probabilities from seen to unseen pairs. An example of an MDP where BCQ's error grows exponentially is shown.

Using a technique similar to Value-Decomposition Networks (Sunehag et al., 2017), authors extend their method to the case of collaborative, multi-agent batch RL.
The method is tested on D4RL: standard single agent batch RL tasks, and Starcraft II for multi-agent evaluation.


**Limitations And Societal Impact:**

Yes

**Main Review:**

1. A big part of the paper is the introduction of the Implicit Constraint. The constraint itself is very similar to the one proposed in [Ghasemipour et al. (2020)](https://arxiv.org/abs/2007.11091) (EMaQ), which uses this update rule:
$$TQ := r + \gamma \mathbb{E}_{s'} \mathbb{E}\_{a^\prime\sim \mu} \big[\text{max}\_{a'\in \text{dataset}} Q(s', a') \big] $$
It feels necessary for the authors to compare ICQ to EMaQ in the text and in the experiments.
2. In particular, I think EMaQ ablates against the exact version of IC in Appendix J. Comparison with Softmax Backup Operators, claiming superior performance of EMaQ. It would be useful if authors clarified differences between these two methods, if any.
3. The single-agent results are presented on little tasks (7, with one difficulty (based on the quality of the behavior's policy) version each, compared to 45 in CQL or 25 in EMaQ).
4. The presentation of results on a plot only, as opposed to a table makes it hard to compare with results reported by other papers, but it looks that at least authors' implementation of CQL doesn't reproduce the scores claimed by CQL's authors.
5. In a number of domains performance of ICQ tracks BC, suggesting that either the domains are too simple, or the performance isn't very impressive.

Assuming that the method of IC was indeed known in the community, the only methodical novelty that the paper presents is the use of VDNs to the case of multi-agent batch RL. The level of innovation there is limited, as the modification of BCQ, etc. is a relatively straightforward application of the ideas from VDN. Given also limited single-agent results and non-comprehensive experiment suite, I suggest rejection.

Minor comments:
- 24: "the accurately estimated values"
- 111: "Let us define"
- 128: "We control... in different datasets to be equal"
- It would be worthwhile to try to bring Fig 4 and 5 closer to the place where they are referred to.

edit: increasing time spent and score to 5 after authors included results on more domains and re-validated previous results of CQL.

**Time Spent Reviewing:**

5

---

> ### Author Response · Authors · 2021-08-10
> **Response to Reviewer 5dG2**
>
> Dear Reviewer,
>
> Thanks for your comments. We hope the following statement clears your concern. We add the supplementary experimental results on single-agent offline tasks as part of our response.
>
> **Q1: The comparison between EMaQ and ICQ**
>
> A1: Although the insights between EMaQ and ICQ have a little similarities, we respectfully disagree that the derived and implement the process of ICQ is very similar with EMaQ, even deny the contribution of the paper for this reason:  ICQ adopts the softmax operator in batch to approximate the partition function Z without behavior policy model $\mu$. However, EMaQ has to learn a behavior policy model $\mu$, which may be unsafe due to the model error. Although EMaQ has compared with softmax backup operators and claiming superior performance of EMaQ, the softmax-version of EMaQ cannot be equivalent with ICQ. Also, in the additional experimental results, we show the better performance of ICQ than EMaQ in complex D4RL tasks.
>
> **Q2: The insufficient experimental results on D4RL tasks.**
>
> A2: We test ICQ with six offline RL methods include EMaQ. Please refer to the supplementary experimental results. ICQ achieves high performance in complex tasks, such as antmaze-medium/large, adroit-human/expert tasks. However, EMaQ has a poor performance in antmaze-medium/large. The results of EMaQ come from [1].
>
> [1] Ghasemipour, Seyed Kamyar Seyed, Dale Schuurmans, and Shixiang Shane Gu. "Emaq: Expected-max q-learning operator for simple yet effective offline and online rl." International Conference on Machine Learning. PMLR, 2021.
>
> **Q3: The performance of CQL.**
>
> A3: We used the code provided by the author in Github without any modification in D4RL tasks.
> However, due to its extremely sensitivity to hyperparameters, CQL has very unstable performances.
>
> To clear your confusion, we re-compare ICQ with CQL in D4RL tasks, where the results for CQL are taken from the original paper. Please refer to the supplementary experimental results. ICQ still works better in complex single-agent offline tasks than CQL, such as antmaze-medium/large, door/hammer-human. We appreciate that you pointed out the confusion for experiments on D4RL, and we have replaced the results of CQL on D4RL with reported results in the CQL paper.

---

> > ### Comment · Reviewer_5dG2 · 2021-08-10
> > **Thank you for more results**
> >
> > Thank you for providing a table with more results. As you say the results for CQL are taken from the paper, I think for human-adroit-relocate result should be 0.35.
> >
> > I think the extra results strengthen your paper. I will increase my score by a point. I am planning on coming back and commenting on comparison with EMaQ after thinking on your response more carefully.

---

> > > ### Author Response · Authors · 2021-08-10
> > > **Thanks for raising the score!**
> > >
> > > Dear Reviewer,
> > >
> > > We really appreciate the feedback from the reviewer, and thanks for raising the score!
> > >
> > > Although the insight of ICQ and EMaQ is don't use unseen pairs, we want to explain their differences further.
> > >
> > > Our most important discovery in offline tasks in the past year is that we should not try to learn the additional behavior model $\mu$ or add the regularization term to avoid the extrapolation error, which is the significant difference between ICQ with other offline RL methods.
> > > The behavior model will induce additional errors, which may be unsafe due to the model error. Also, it is hard to balance the regularization term with other optimization goals. Therefore, we propose ICQ, which re-weights the target-Q and actor-loss by importance sampling. The solid performance and simple form of ICQ lays a foundation in multi-agent offline tasks.
> > >
> > > The other difference between ICQ and EMaQ is that ICQ needs to learn one Q function and one target Q function. However, EMaQ has to learn several Q functions to combat overestimation bias.
> > >
> > > We admire EMaQ very much, and we do think ICQ is groundbreaking based on EMaQ. We will discuss the difference between ICQ and EMaQ in the critical place of the paper.
> > >
> > > Best regards!
> > >
> > > Authors

---

> > > > ### Comment · Reviewer_5dG2 · 2021-08-10
> > > > **Thanks for clarifications**
> > > >
> > > > After looking at the EMaQ more carefully, I agree that the two methods are different, yet based on the same high-level idea of only using the dataset actions for bootstrapping. I will appreciate including the analysis from your previous comment in the paper, as it clarifies the difference between the two methods.
> > > >
> > > > The updated results are mixed, in particular in poor-quality datasets with a weak behavioral policy. Because of that, I find your argument of not modeling $\mu$ due to extrapolation error weak, as ICQ only does well (better than CQL/EMaQ) on the datasets with a good behavioral policy, where the extrapolation error will not be as big of a problem as elsewhere. I rather see your method trading off performance in harder (further from the bp) tasks for better scores on the easier ones, than "significantly and constantly outperform[ing] baselines in performance and sample efficiency", as presented.
> > > >
> > > > I am thus keeping my weak rejection recommendation.

---

> > > > > ### Author Response · Authors · 2021-08-12
> > > > > **Thanks for the response!**
> > > > >
> > > > > Thanks for the valuable comments and suggestions!

---

### Official Review · Reviewer_xuSQ · 2021-07-16

**Rating:** 6
**Confidence:** 3

**Summary:**

The main contribution of the paper is to investigate multi-agent offline RL settings and highlighting the additional challenges in this settings compared to single-agent offline RL. Moreover, the paper proposes ICQ for offline RL, which combines importance sampling based off-policy approach with KL-constraint. ICQ empirically leads to good performance on challenging multi-agent offline tasks (StarCraft II) where existing offline RL approaches are reported to fail.

**Limitations And Societal Impact:**

Importance sampling based approaches are known to be fragile especially in the cases when the distribution of the learned policy is very different from the behavior (or data) distribution. I expect ICQ to suffer from similar issues and request the authors to clarify if they think otherwise. For example, [1] argues that he maximum improvement that can be reliably obtained via importance sampling is limited by (i) the suboptimality of the behavior policy; (ii) the dimensionality of the state and action space; (iii) the effective horizon of the task.  See the discussion in [1] for more details.

Appendix F does a good job at mentioning other limitations in the appendix.

[1] Levine, Sergey, et al. "Offline reinforcement learning: Tutorial, review, and perspectives on open problems." arXiv preprint arXiv:2005.01643 (2020).

--------------------------------------------------------------------------------------------------------------------------------------------------------------------------------
*Update*: Based on author response, I weakly recommend acceptance (updating my score to 6) despite the poor clarity of the initial version of the paper. I strongly advise the authors to revise the paper to clarify the relationship to BRAC and KL-control, include additional experiments on harder D4RL tasks, fixing results for their CQL baselines (mention of hyperparameter sensitivity of CQL) and a more thorough discussions of the limitations (including the ones pointed by xuSQ in their response).

**Main Review:**

**Originality**: The submission seems to explore multi-agent settings for offline RL which haven't received much attention from the community. Furthermore, exponential error increase with the number of agents seem to be an insightful observation. That said, the proposed method ICQ, seems to fall under the broad category of methods which apply KL-divergence constraints, which has also been applied in the context of offline RL, including BRAC[1] and KL-control [2].

**Quality**: The submission seems to be theoretically sound as well as the claims seem to be supported by theoretical justifications in tabular settings (albeit with simplified assumptions). However, it is unclear to me whether the comparison to other methods is fair. I have some questions regarding the empirical section :
- How were the hyperparameters selected for the baselines vs ICQ? It seems the hparams correspond to the default value used in prior work but those values correspond to different set of tasks.
- Why is the reported CQL performance so poor on the D4RL tasks -- CQL was claimed to be much better than BCQ on a variety of D4RL tasks including AntMaze and Kitchen tasks. This seems to indicate either a bug in the CQL implementation or a poor selection of hyperparameters.
- Since BRAC and KL-control are more closer to ICQ, could the authors elaborate on their choice to instead use BCQ as their baseline?
- What if I had a dataset only containing just poor/medium performance behavior policies only? This experiment might empirically address the limitation of using importance sampling in ICQ (see limitation section below).


**Clarity**: This paper as written currently is often not easy to parse and could be improved.
- One major issue that sufficient intuition is not provided to the reader before the theoretical results are presented, for example, what are the implications of Theorem 1 and 2? Why would standard offline RL approaches are expected to fail in lieu of Theorem 2?
- While I appreciate the details of derivation included in section 4.1 for the loss function, these details make it harder to read that section and could be presented more clearly.  One would be to work backwards from the final loss to its derivation, so that the reader is aware of what is being derived. Same issue exists with Theorem 4.

**Significance**: It is likely that this paper might generate more interest in the multi-agent offline RL settings. It is hard to say if future work would build on top of ICQ as it is unclear if the baselines were implemented correctly and the comparisons were fair.



[1] Wu, Yifan, George Tucker, and Ofir Nachum. "Behavior regularized offline reinforcement learning." arXiv preprint arXiv:1911.11361 (2019).

[2] Jaques, Natasha, et al. "Way off-policy batch deep reinforcement learning of implicit human preferences in dialog." arXiv preprint arXiv:1907.00456 (2019).

**Time Spent Reviewing:**

8

---

> ### Author Response · Authors · 2021-08-10
> **Response to Reviewer xuSQ**
>
> Dear Reviewer,
>
> Thanks for finding our theoretical results sound. In the following, we will address issues that you are concerned about. We hope that you find the following statement improving the paper and making the paper clear. We add supplementary experimental results on single-agent offline tasks as part of our response.
>
> **Q1: (Originality) Does the ICQ fall under the broad category of methods that apply KL-divergence constraints like BRAC?**
>
> A1: Although both ICQ and BRAC use the KL-divergence constraints, we respectfully disagree that ICQ falls under the broad category of BRAC for two reasons:
>
> 1) ICQ focuses on avoiding using unseen pairs in policy evaluation by re-weighting the target Q using an importance sampling ratio. The importance sampling ratio is derived by maximizing Q-values, subject to a KL-constraint between current policy $\pi$ and the behavior policy $\mu$. However, BRAC aims to encourage the learned policy to be close to the behavior policy by adding a regularization term. Therefore, the most significant difference between ICQ and BRAC is: ICQ can eliminate the extrapolation error to zero theoretically based on the analysis in Theorem 1. In contrast, BRAC eliminates the extrapolation error empirically.
>
> 2) By applying the constraint implicitly, ICQ avoids an additional regularization term or even training a separate network as BCQ does, which contributes ICQ to be a simple yet efficient method. In contrast, BRAC has to balance the additional regularization term and other optimization goals.
>
> **Q2: (Quality) Why is the reported CQL performance so poor on the D4RL tasks?**
>
> A2: We used the code provided by the author in Github without any modification in D4RL tasks. However, due to extreme sensitivity to hyperparameters, CQL has very unstable performances.
>
> To clear your confusion, we re-compare ICQ with CQL in most D4RL tasks, where the results for CQL are taken from the original paper. Please refer to the supplementary experimental results. ICQ still works better even in complex single-agent offline tasks than CQL, such as antmaze-medium/large, door/hammer-human. We appreciate that you pointed out the confusion for experiments on D4RL, and we have replaced the results of CQL on D4RL with reported results in the CQL paper.
>
> **Q3: (Quality) Why not use BRAC and KL-control as baselines? Why choose BCQ as baselines?**
>
> A3: A1 explains the difference between ICQ and BRAC. Unlike BRAC, BCQ adopts a generative model to avoid unseen pairs. However, the generative model may be unsafe due to the model error. BCQ-MA demonstrates that the multi-agent offline tasks are less tolerant of extrapolation errors than single-agent offline tasks.
>
> Further, in the supplementary experimental results, we compare ICQ with BRAC-p and BRAC-v, where the results of BRAC are taken from [1]. In adroit-expert/human, antmaze-medium/large, ICQ achieves high performance while the performance of BRAC is so poor. Moreover, BRAC just achieves sound performance in mujoco tasks, which demonstrates well the statement in A1.
>
> [1] Fu, Justin, et al. "D4rl: Datasets for deep data-driven reinforcement learning." arXiv preprint arXiv:2004.07219 (2020).
>
> **Q4: (Quality) How to select hyperparameters for the baselines? If the comparisons were fair?**
>
> A4:
> We believe our comparison to the BCQ and CQL methods is fair for two reasons: 1) we have fine-tuned hyperparameters of BCQ-MA and CQL-MA in StarCraft II. 2) we adopt the same neural networks structure and value-decomposition assumption as ICQ-MA. However, we find it is not enough to improve the performance of BCQ-MA and CQL-MA by fine-tuning hyperparameters.
>
> It is not surprising that BCQ-MA and CQL-MA have poor performance in the multi-agent experimental results. In the supplementary experimental results, current offline methods are still not good enough in handling the extrapolation error in complex single-agent offline tasks, such as antmaze-medium/large and adroit-human. Based on the analysis in Section 3 and the toy example in Figure 2 in the paper, the extrapolation error is quickly accumulated with the growth of the number of agents. Multi-agent offline tasks are less tolerant of extrapolation errors than single-agent offline tasks. Therefore, most single-agent offline RL methods cannot be extended in multi-agent offline tasks.
>
> In contrast, ICQ alleviates the extrapolation error by a simple yet efficient method instead of training a separate network as BCQ or using additional regularization terms, such as CQL or BRAC.  As evidence, ICQ achieves high performance in complex single-agent offline tasks such as antmaze and adroit. The state-of-the-art performance of ICQ on single-agent offline tasks lays a solid foundation for multi-agent offline tasks.
>
> **Q5: (Quality) How to deal with dataset only containing just poor/medium performance behavior policies?**
>
> A5: As we adopt the constraint with the learned policy and the behavior one, ICQ works well in the high-quality dataset theoretically and maybe has a limited performance with randomly sampled data. It can be seen that ICQ works well in mujoco-medium(or medium-expert), while ICQ has a poor performance in mujoco-random(or medium-replay). Nevertheless, we still regard ICQ as a good choice in multi-agent and complex single-agent tasks compared with current offline RL methods.
>
> **Q6: (Clarity) What are the implications of Theorem 1 and 2? Why would standard offline RL approaches are expected to fail in lieu of Theorem 2?**
>
> A6: We analyze the extrapolation error propagation in single-agent and multi-agent offline tasks based on Theorem 1 and Theorem 2. Based on these analyses, we propose our algorithms ICQ and ICQ-MA.
>
> We demonstrate the extrapolation error is accumulated quickly in multi-agent offline tasks in Theorem 2. Multi-agent offline tasks are less tolerant of extrapolation errors than single-agent offline tasks. Therefore, most single-agent offline RL methods cannot be extended in multi-agent offline tasks.
>
> **Q7: (Clarity) Work backward from the final loss to the loss derivation.**
>
> A7: Thanks for your detailed suggestions and we also apologize for the confusion. We have improved the current version as per your suggestions.
>
> Thanks again for the valuable comments. We hope our additional experimental results and explanation have cleared the concern.  We sincerely hope that the reviewer can re-evaluate our paper after seeing our response. More comments on further improving the presentation are also very much welcomed.

---

> ### Author Response · Authors · 2021-08-10
> **Thanks for raising the score!**
>
> We appreciate the reviewer for raising the score to 6! Thanks for the valuable comments and suggestions!

---

### Author Response · Authors · 2021-08-10
**General remark (with supplementary experimental results)**

We first thank all the reviewers for their constructive and valuable comments. We have noticed that many of the reviews are quite positive, finding our work “interesting” and “theoretical results sound” in general. We really appreciate those positive comments. It seemed a pity that most of the concerns came from our insufficient experiments in single-agent offline tasks and insufficient explanation of the algorithm.  In supplementary experimental results, we test ICQ on single-agent offline tasks.  We add more detailed explanations on the ICQ for each reviewer.  We sincerely hope that our response can clear reviewers' concerns.  More discussions and suggestions on further improving the paper are also always welcomed!

| Dataset type | Environment | ICQ (ours) | Estimation Error | BRAC-p | BRAC-v | BCQ | CQL | EMaQ | MOPO |
| ----------- | ----------- | :---: | :---: | :---: | :---: | :---: | :---: | :---: | :---: |
| expert | adroit-door | **102.9$\pm$3.6** | 0.64$\pm$0.23  | -0.3 | -0.3 | 99.0 | - | - | - |
| expert | adroit-relocate | **107.2$\pm$5.2** | 1.36$\pm$0.87 | -0.3 | -0.4 | 41.6 | - | - | - |
| expert | adroit-pen | **115.8$\pm$2.1** | 6.67$\pm$1.31 | -3.5 | -3.0 | 114.9 | - | - | - |
| expert | adroit-hammer | **124.3$\pm$2.5** | 16.05$\pm$2.48 | 0.3 | 0.3 | 107.2 | - | - | - |
| human | adroit-door | 6.4$\pm$2.4 | 0.40$\pm$0.35 | -0.3 | -0.3 | -0.0 | **9.1** | - | - |
| human | adroit-relocate | **1.5$\pm$0.7** | 1.87$\pm$0.80 | -0.3 | -0.3 | -0.1 | 0.35 | - | - |
| human | adroit-pen | **90.6$\pm$1.3** | 17.62$\pm$12.37 | 8.1 | 0.6 | 68.9 | 55.8 | - | - |
| human | adroit-hammer | 0.2$\pm$0.9 | 0.17$\pm$0.29 | 0.3 | 0.2 | 0.5 | **2.1** | - | - |
| fixed | antmaze-umaze | 85.0$\pm$2.7 | -0.03$\pm$0.01 | 50.0 | 70.0 | 78.9 | 74.0 | **91.0** | - |
| play | antmaze-medium | **80.0$\pm$1.3** | -0.01$\pm$0.02 | 0.0 | 0.0 | 0.0 | 61.2 | 0.0 | - |
| play | antmaze-large | **51.0$\pm$4.8** | -0.02$\pm$0.01 | 0.0 | 0.0 | 6.7 | 15.8 |  0.0 | - |
| diverse | antmaze-umaze | 55.0$\pm$3.3 | -0.03$\pm$0.01 | 40.0 | 70.0 | 55.0 | 84.0 | **94.2** | - |
| diverse | antmaze-medium | **65.0$\pm$3.9** | -0.02$\pm$0.02 | 0.0 | 0.0 | 0.0 | 53.7 | 0.0 | - |
| diverse | antmaze-large | **44.0$\pm$4.2** | -0.02$\pm$0.01 | 0.0 | 0.0 | 2.2 | 14.9 | 0.0 | - |
| medium | mujoco-walker2d | 71.8$\pm$0.7 | 0.23$\pm$0.21 | 77.5 | **81.1** | 53.1 | 79.2 | 80.5 | 17.8 |
| medium | mujoco-hopper | **58.6$\pm$2.7** | 0.25$\pm$0.04 | 32.7 | 31.1 | 54.5 | 58.0 | 53.3 | 28.0 |
| medium | mujoco-halfcheetah | 42.5$\pm$0.3 | 0.12$\pm$0.12 | 43.8 | **46.3** | 40.7 | 44.0 | 47.7 | 42.3 |
| medium-expert | mujoco-walker2d | **98.9$\pm$0.2** | 0.20$\pm$0.03 | 76.9 | 81.6 | 57.5 | 98.7 | 91.4 | 44.6 |
| medium-expert | mujoco-hopper | 109.0$\pm$0.6 | 0.18$\pm$0.01 | 1.9 | 0.8 | 110.9 | 110.0 | **114.4** | 23.7 |
| medium-expert | mujoco-halfcheetah | **110.3$\pm$0.1** | 0.12$\pm$0.23 | 44.2 | 41.9 | 64.7 | 104.8 | 99.0 | 63.3 |
| random | mujoco-walker2d | 6.5$\pm$0.1 | -1.3$\pm$0.2 | -0.2 | 1.9 | 4.9 | 7.0 | 6.7 | **13.6** |
| random | mujoco-hopper | 9.9$\pm$0.8 | 0.83$\pm$0.05 | 11.0 | **12.2** | 10.6 | 10.8 | 9.6 | 11.7 |
| random | mujoco-halfcheetah | 2.2$\pm$0.0 | -0.43$\pm$0.05 | 24.1 | 31.2 | 2.2 | **35.4** | 24.7 | **35.4** |
| medium-replay | mujoco-walker2d | 10.2$\pm$1.6 | 0.31$\pm$0.18 | -0.3 | 0.9 | 15.0 | 26.7 | 13.0 | **39.0** |
| medium-replay | mujoco-hopper | 15.6$\pm$2.2 | 0.40$\pm$0.21 | 0.6 | 0.6 | 33.1 | 45.3 | 46.3 | **67.5** |
| medium-replay | mujoco-halfcheetah | 28.0$\pm$0.3 | 0.79$\pm$0.12 | 45.4 | 47.7 | 38.2 | 46.2 | 34.5 | **53.1** |

Table 1. Performance of ICQ with six offline RL baselines on the single-agent offline tasks with the normalized score metric proposed by D4RL benchmark [1], averaged over three random seeds with standard deviation.
Scores roughly range from 0 to 100, where 0 corresponds to a random policy performance, and 100 indicates an expert.
Estimation error refers to the average estimated Q-values minus the average returns.
The results for BRAC-p, BRAC-v, BCQ are taken from [1].
The results for CQL are taken from [2].
The results for EMaQ are taken from [3].
The results for MOPO are taken from [4].

[1] Fu, Justin, et al. "D4rl: Datasets for deep data-driven reinforcement learning." arXiv preprint arXiv:2004.07219 (2020).

[2] Kumar, Aviral, et al. "Conservative q-learning for offline reinforcement learning." arXiv preprint arXiv:2006.04779 (2020).

[3] Ghasemipour, Seyed Kamyar Seyed, Dale Schuurmans, and Shixiang Shane Gu. "Emaq: Expected-max q-learning operator for simple yet effective offline and online rl." International Conference on Machine Learning. PMLR, 2021.

[4] Yu, Tianhe, et al. "Mopo: Model-based offline policy optimization." arXiv preprint arXiv:2005.13239 (2020).

---

### Decision · Program_Chairs · 2021-09-28

**Decision:**

Accept (Spotlight)

**Comment:**

In the light of authors' response and additional experiments, some of the reviewers decided to increase their score but they are of the opinion that the paper could be improved in terms of presentation clarify. The additional results provide extra evidence for the method but they require a set of further edits which warrant another round of reviews. In addition, some of the reviewers are not fully convinced by the interpretation of the results by the authors. I suggest that the authors address the reviewers' points, include the extra results and improve the presentation of the paper and resubmit.

**Consistency Experiment:**

NeurIPS has a long history of experimentation. In 2014, NeurIPS ran an experiment in which 10% of submissions were reviewed by two independent committees to quantify the randomness in the review process. This year, we repeated a variant of this experiment to see how the quality of the review process has changed over time.  This paper was part of the experiment and was therefore assigned to two committees (consisting of reviewers, an Area Chair, and a Senior Area Chair) that reached independent decisions.  If both committees made the same recommendation, this recommendation was followed. If a single committee recommended acceptance, the paper was accepted (with the exception of a few cases in which the other committee identified what we considered a fatal flaw, e.g., an error in a key result).

This copy’s committee reached the following decision: **Reject**

The other committee assigned to the paper recommended **Accept (Spotlight)**.  You can find the other set of reviews, along with any follow up discussion with the authors here:
https://openreview.net/forum?id=yNzF41lHYV